# Current Trends in the Utilization of Photolysis and Photocatalysis Treatment Processes for the Remediation of Dye Wastewater: A Short Review

S M Anisuzzaman [1], Collin G. Joseph [2,3,4,*], Chuan Kian Pang [2,4], Nur Ammarah Affandi [2,4], Sitti Nurazida Maruja [2,4] and Veena Vijayan [2,4]

1   Chemical Engineering Programme, Faculty of Engineering, Universiti Malaysia Sabah, Kota Kinabalu 88400, Sabah, Malaysia; anis_zaman@ums.edu.my
2   Sonophotochemistry Research Group, Faculty of Science and Natural Resources, Universiti Malaysia Sabah, Kota Kinabalu 88400, Sabah, Malaysia; kian8808@gmail.com (C.K.P.); ms1821030t@student.ums.edu.my (N.A.A.); bs18110574@student.ums.edu.my (S.N.M.); veenavijayan@ums.edu.my (V.V.)
3   Water Research Unit, Faculty of Science and Natural Resources, Universiti Malaysia Sabah, Kota Kinabalu 88400, Sabah, Malaysia
4   Industrial Chemistry Programme, Faculty of Science and Natural Resources, Universiti Malaysia Sabah, Kota Kinabalu 88400, Sabah, Malaysia
*   Correspondence: collin@ums.edu.my; Tel.: +60-88-320000 (ext. 2117)

**Abstract:** Development in the textile industry leads to an increased demand for the use of various dyes. Moreover, there is the use of some dyes in the food industry as well as medical diagnostics. Thereby, increased demand for dyes in various fields has resulted in dye-containing wastewater. Only a small portion of the generated wastewater is adequately treated. The rest is usually dumped or otherwise directly discharged into the sewage system, which ultimately enters rivers, lakes, and streams. The handling and disposal of such concentrated wastewater, especially the dye-containing wastewater, is considered to be a major environmental issue from the moment of its generation to its ultimate disposal. Conventional water treatment methods such as flotation, filtration, adsorption, etc., are non-destructive physical separation processes. They only transfer the pollutants to other phases, thereby generating concentrated deposits. The advanced oxidation process (AOP) is one of the most effective emerging methods for the treatment of wastewater containing chemical pollutants. The method involves the formation and interaction of highly reactive hydroxyl radicals under suitable activation conditions. These radicals are non-selective and efficient for the destruction and eventual mineralization of recalcitrant organic pollutants. This review aims at the pros and cons of using photocatalysis as an efficient AOP to degrade dye-containing wastewater.

**Keywords:** photolysis; photocatalysis; photodegradation; degradation efficiency; UV irradiation

## 1. Introduction

Dyes can be described as the substances used to impart color to a given substrate in various industries such as silk, paper, and paints [1]. Dyes can be classified in multiple ways based on industrial applications and by their chemical types or chromophore structures [2]. Dyes are characterized by the capacity to absorb light radiation in the visible spectrum, which is the wavelength from 380 to 750 nm [1]. Dyes are organic compounds consisting of two groups in their molecules, which are the chromophore and the auxochrome [3]. The most common chromophores of dyes are azo (-N=N-), nitro (-NO$_2$), nitroso (-N=O), thiocarbonyl (-C=S), and alkenes (-C=C-) [3]. The chromophore is the active site of the dyes, which is the absorbing part of the visible spectrum. The electronic properties of the chromophore result in the color imparted by the dyes [1]. The auxochrome in the

chromogenic molecules may intensify the color of the dye and acidic groups such as $SO_3$ and $COOH$ are frequently introduced into dye structures to enhance their solubility [4].

Table 1 shows the classification of dyes based on their applications. Dyes also can be divided into cationic, anionic, and non-ionic dyes. Cationic dyes have cationic functional groups that may dissolve into positively charged ions in an aqueous solution, such as Methylene blue (MB), Rhodamine B (RhB), Malachite green (MG), Rhodamine 6G (Rh6G), and Crystal violet (CV). Because the onium group is the most frequent cationic functional group, most of the cations are $N^+$ ions. Anionic dyes include Acid orange 7 (AO7), Eosin Y (EY), Methyl orange (MO), Acid red 14 (AR14), Alizarin red S (ARS), Rose Bengal (RB), Phenol red (PR), etc. Anionic functional groups are found in all anionic dyes [5].

**Table 1.** Classification of dyes based on industrial application.

| Industrial Application Class | Applications | Chemical Type/Chromophore Structure | Example |
|---|---|---|---|
| Disperse dye (non-ionic) | Polyester, nylon, cellulose, cellulose acetate, acrylic fibers, polyamide, plastics | Azo, nitro, styryl, anthraquinone, benzodifuranone, | Disperse violet 26 (DV26), Disperse blue 27 (DB27) |
| Direct dye (anionic) | Paper, cellulose fibers, nylon, rayon, cotton, viscose, leather | Azo, oxazine, thiazole, stibene, phthalocyanine | Direct red 28 (DR28), Direct orange 26 (DR26) |
| Reactive dye (anionic) | Cellulose fibers, silk, cotton, wool fibers, nylon | Anthraquinone, formazan, oxazine, phthalocyanine, azo, triphenylmethane | Reactive blue 19 (RB19), Reactive blue 5 (RB5) |
| Vat dye (non-ionic) | Cellulose fibers, cotton, viscose, wool | Anthraquinone, Indigoid | Vat blue 1 (VB1), Vat blue 4 (VB4) |
| Basic dye (cationic) | Acrylic, ink, paper, silk, wool, cotton, treated nylon, modified polyester, polyacrylonitrile | Triarylmethane, azo, xanthene, Triphenylmethane, hemicyanine, cyanine, acridine, diazahemicyanine, anthraquinone, oxazine, thiazine | Basic blue 6 (BB6), MB, MG |
| Acid dye (anionic) | Nylon, wool, leather, food, silk, cotton, cosmetics, ink-jet printing, paper, modified acrylics | Anthraquinone, xanthene, azo, nitrodiphenylamine, triphenylmethane, nitroso, azine, nitro, indigoid | AO7, Acid yellow 36 (AY36) |

Source: Morsy et al. [2]; Hunger [6]; Katheresan et al. [7].

Ensuring the availability of clean water is essential for humans and terrestrial and aquatic animals and plants. In compliance with the sustainable development goals (SDG) to ensure this availability in the pursuit of global economic growth and industrial developments, generated wastewater has to be treated and remediated for reuse. Textile industries comprise one of the most labor-intensive industries, providing employment to various downstream and upstream sectors. At the same time, this industry produces a large amount of dye-contaminated wastewater that is released into rivers and streams. Modern dyes are synthetically designed to withstand weathering processes or biodegradation. Over the last few decades, researchers and scientists have investigated new techniques and methods to treat and remediate dye-contaminated wastewater. This review paper explores the current and updated trends in this field with a focus on photolysis and photocatalysis treatment processes.

*Occurrence of Dyes in the Aquatic Environment*

In the last few decades, the contamination of water by dyes has continuously existed due to the release of dyes from industrial textile discharge. In Malaysia, the textile and dyestuff industry is one of the developing industries that contribute to the country's economic growth. However, these dyestuff industries require high water consumption, which produces a massive quantity of pollutants and is often disposed into water bodies with

little to no pre-treatment, leading to critical environmental problems [8]. The wastewater discharged from the dyestuff industries has been considered the primary source of water contamination and pollution [9]. Owing to the presence of aromatic rings and reactive ring groups such as sulfur and naphthol along with heavy metals such as chromium, lead, copper, cadmium, and mercury, textile wastewater is regarded as highly toxic [4].

The basic textile processing technology includes desizing, scouring, bleaching, mercerizing, and the dyeing process. The different dyes, inorganic and organic-based compounds used in the wet processing, etc., will influence the wastewater components [10]. Wet processes often use many chemicals and large quantities of water; 1 kg (kg) of fabric requires around 80–150 $m^3$ of water [11]. Table 2 shows the wet processes producing textile wastewater in which the dyeing, printing, and finishing steps give the most contribution to fluctuations in water quality parameters such as pH, temperature, COD, BOD, color, and salinity.

**Table 2.** Textile wastewater produced by wet processes.

| Process | Quality of Textile Effluent |
| --- | --- |
| Desizing | High TSS, high BOD, neutral pH |
| Scouring | High TTS, high BOD, high alkalinity, high temperature |
| Bleaching, Mercerizing | High TSS, high BOD, alkaline wastewater |
| Heat-setting | Low TSS, low BOD, alkaline wastewater |
| Dyeing, printing, and finishing | High TSS, BOD, COD, wasted dyes, neutral to alkaline wastewater |

Source: Naveed and Bhatti, [12].

## 2. Dye-Removal Methods

Dyes are noticeable even in small amounts and present as more aesthetically displeasing in contaminated wastewater bodies [6]. Moreover, many dyes are hazardous for aquatic organisms and humans. Hence, the interest in the treatment of dye wastewater for reuse has been great due to the shortage of clean, natural water sources [2]. An effective method is a must to remove large amounts of dyes rapidly, cost-effectively, and without producing secondary contaminants [7]. Conventional dye removal methods have been classified into three main categories, which are physical, chemical, and biological methods.

### 2.1. Physical

Physical methods are primarily used to separate large dissolved matter and to recover and reuse valuable substances used in the main processes [13]. Many physical technologies have been applied to the removal of dyes, such as ion exchange, adsorption, and the membrane filtration method [14]. The coagulation or flocculation process is primarily utilized for the effective removal of dispersed dyes. However, it has the downside of increasing sludge formation volume [2,15]. This treatment method is commonly applied as a pre-treatment step for the removal of dyes from effluents [16]. Filtration methods such as reverse osmosis, microfiltration, and nanofiltration are used to remove the dyes from water for reuse, but this technique is not cost-effective, as it requires high maintenance costs [2]. Another physical method is the adsorption, which is more efficient for the decolorization of dyes than the coagulation method. It uses low-cost adsorbents such as polymeric resins and bentonite clay, but it is not cost-efficient as the adsorbents are usually used once without being able to reuse [2].

### 2.2. Biological

Biological treatment is a widely used technique in wastewater treatment that has been used for over 150 years [6]. This technique is based on the activated sludge that contains a variety of aerobic and anaerobic microorganisms such as bacteria and green algae [4]. The biological method is preferred over physical and chemical practices as no

hazardous by-products are produced, and it is a more cost-effective method on an industrial scale [2]. Laccase is one of the examples of enzymes of interest for dye removal due to its physicochemical properties as a promising biocatalyst. The mechanism of laccase on dye degradation is by catalyzing three types of reaction: the direct oxidation of phenolic substances, the indirect oxidation of non-phenolic substrates, and coupling reactions with reactive intermediate radicals formed during direct oxidation [2]. However, despite the advantages of the biological method, its effectiveness for dye removal is limited since many dyes are not biodegradable and are toxic to microbes as synthetic dyes are designed in such a manner.

*2.3. Chemical*

The chemical method of dye removal can be classified into two categories, which are conventional chemical treatments and chemical oxidation processes [17]. In conventional chemical treatment methods, treatments such as adsorption using various adsorbents, coagulation, and flocculation methods and electrochemical methods merely transform the organic pollutants from one phase to another, i.e., causing secondary pollution [6]. The advantages of the conventional chemical method are lower sludge generation and high pollutant removal efficiency, while high operational and chemical costs and secondary waste are some of the disadvantages of chemical treatment methods [4]. Among the chemical methods, oxidation is effective and practical for large-scale wastewater treatment [18]. Chemical oxidation degrades organic molecules by using powerful oxidizing chemicals such as ozone, hydrogen peroxide, chlorine, and potassium permanganate [6]. The fundamental chemical oxidation process happens naturally when air and oxygen are present, but it is insufficient for heavily contaminated wastewater. Hence, there is a significant necessity to develop methods that can remove the pollutants effectively.

## 3. Advanced Oxidation Processes (AOPs)

In recent years, advanced oxidation processes (AOPs), which use combinations of ultraviolet irradiation, catalysts, and oxidants to produce hydroxyl radicals ($^\bullet$OH) in solutions have received attention as a technology for the degradation of harmful organic compounds in wastewater. AOPs were first proposed for the treatment of drinking water in the 1980s and later AOPs have been utilized for the treatment of various types of wastewaters [19]. Due to the complicated aromatic structure and resistant nature of dyes and compounds, conventional biological and chemical oxidation approaches are ineffective for the degradation of these compounds [20]. AOPs are viewed as a highly competitive technology in water treatment for removing organic pollutants that are not biodegradable and inactivating pathogenic microorganisms that cannot be treated with traditional techniques [21]. AOPs are mainly used to destroy organic contaminants in water and sewage [19].

In order to identify the formation as well as the reaction mechanism of reactive oxygen species (ROS) in each of the AOP systems, it is critical to study the principles and the degradation mechanism of organic pollutants under these AOPs. The presence of ROS is termed as responsible for the effective mineralization of various organic pollutants under different AOPs. ROS can be produced by external energy sources such as photo, sono, electro, etc., in the presence and absence of catalysts as well as secondary oxidants. AOPs are the oxidation processes that require the rapid generation of hydroxyl radicals ($^\bullet$OH), which is a highly reactive oxygen species, in sufficient quantities to allow the organic contaminants to be oxidized and mineralized by the free radicals to mineral salts, water, and carbon dioxide [22]. Hydroxyl radicals are non-selective in nature and can react with a wide variety of pollutants without any other additives, with rate constants typically in the order of $10^6$ to $10^9$ mol. l.s.$^{-1}$ [13]. AOPs start with the generation of OH radicals which then attack the targeted pollutants [22]. After the process of AOP methods, new oxidized intermediates are produced with lower molecular weights, or in the case of complete mineralization, carbon dioxide and water are produced [13].

AOPs can be classified based on homogenous and heterogenous reactions and based on the presence of irradiation (Figure 1). Homogenous reactions occur in a single phase. Meanwhile, a heterogenous reaction is when the reactants are components of two or more phases, in which, in AOPs, the pollutant is liquid and the catalyst is solid [23–25]. There are many developed methods of AOPs for wastewater treatment processes, as the basic chemical oxidation process is not enough for severely contaminated wastewater [18]. The literature has shown that hydroxyl radicals with high oxidizing potential can be generated through different types of AOPs for the degradation of dyes such as photocatalysis [26], photolysis, UV/$H_2O_2$ [27], photolysis, UV/$O_3$ [28], photo-Fenton [29], electrochemical oxidation [30], ozonation [31], and sonolysis [32]. The benefits of AOPs include rapid reaction rates and non-selective oxidation, which allows multiple pollutants to be treated at the same time and can reduce the toxicity of the pollutants. Mineralization can be completed by AOPs, but it can be costly, thus biological treatment is preferable for the final stage of the treatment of dyes [33].

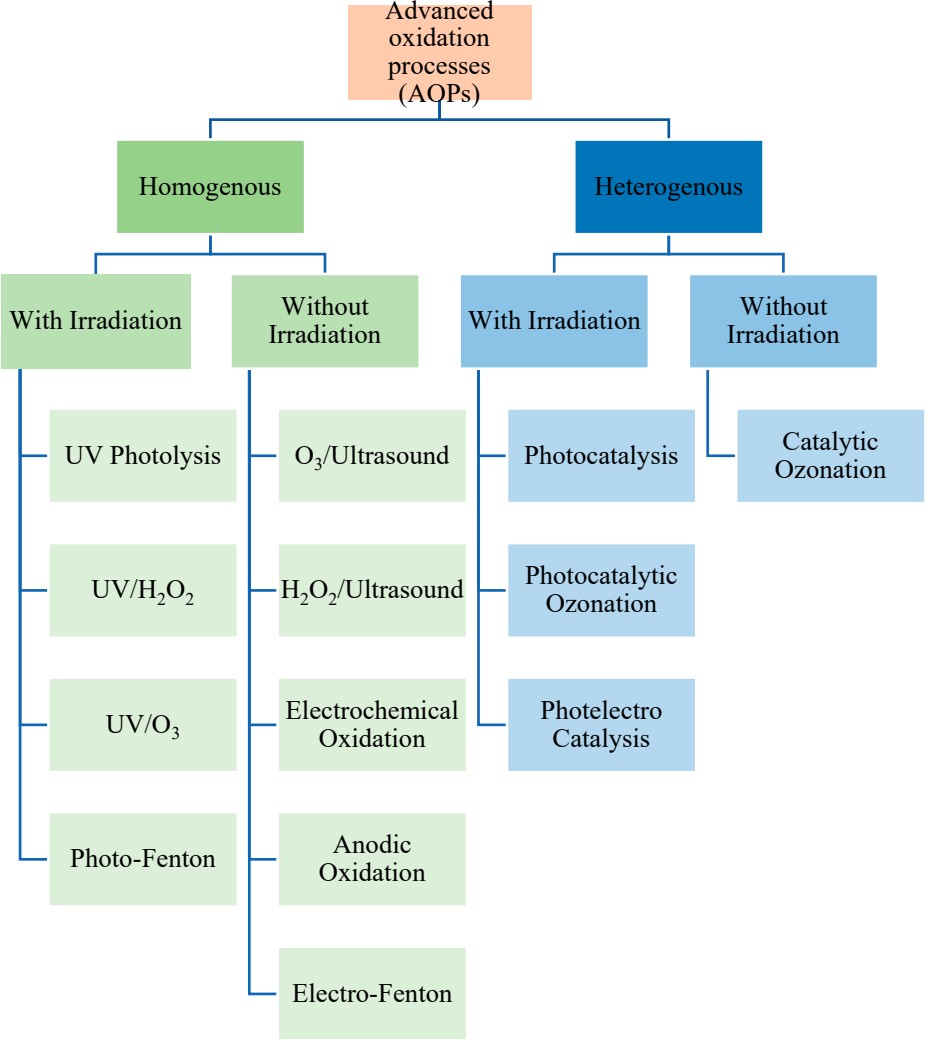

**Figure 1.** Different AOP-based treatment methods. Source: Khan et al. [34].

## 4. Photochemical Treatment of Dyes

Photodegradation is one example of an advanced oxidation process (AOP). This method is the chemical breakdown of a large molecule into non-toxic molecules in which the process of hydroxyl radicals initiated by photons absorbed in the wavelength of UV, visible, and infra-red (IR) spectral ranges, with or without the presence of a catalyst or oxidants [19,34]. Hydroxyl radical generation can be accelerated by the combinations of some commonly

used oxidants. The photodegradation of dyes is considered the most promising technology for the treatment of industrial wastewater processes as it is environmentally friendly, low-cost, and does not cause any secondary pollution [35]. The techniques available for the photodegradation of dyes are UV photolysis, photooxidation by oxidizing agents including hydroxyl radicals, hydrogen peroxide, ozone, and photocatalytic UV oxidation [34].

*History*

Humans first used photochemistry in 1500, when Canaanite peoples arrived along the Mediterranean's eastern shore. The current organic photochemistry stage began in 1866 when Russian scientist Carl Julius von Fritzche found that when a concentrated anthracene solution was subjected to UV light, it precipitated. This precipitation occurs as a result of anthracene molecules joining together in pairs, or dimers, which are no longer soluble. Scientists have found that the materials (dyes and phosphors) must have the capability to absorb optical radiation. Photocatalysis studies using $TiO_2$ have been reported since the early 20th century. In 1938, the photodegradation of dyes by $TiO_2$ was reported. UV absorption produces active oxygen species on the $TiO_2$ surface, causing the photodegradation of dyes. It was also discovered that $TiO_2$ did not alter throughout the photoreaction, despite the fact that the photocatalyst terminology was not used for $TiO_2$ in the paper and it was instead referred to as a photosensitizer [36].

## 5. Current Developments in UV Radiation Sources

Ultraviolet (UV) radiation is in the electromagnetic spectrum that human eyes cannot see [23–25]. There are three classifications of UV radiation based on the interaction of UV radiation with biological materials, which are UV-A (400–315 nm), UV-B (315–280 nm), and UV-C (280–100 nm) [23–25]. UV-A, also known as black light, is related to skin aging and has recently been linked to skin cancer in humans and animals. UV-B is also associated with the risk of skin cancer and it contributes to erythema (sunburn), but in contrast, it is also important in the synthesis of vitamin D, which suggests that it may potentially reduce the risk of prostate, breast, and colon cancer and is also used in phototherapy for baby jaundice [37]. UV-C's effects on organisms are negligible, as it is completely filtered out by the Earth's atmosphere, but UV-C is valuable as a research tool and a sterilizing procedure due to its bactericidal properties at wavelengths of 260–280 nm [25,37]. UV light is used in these photochemical techniques due to the greater energy of their photons, as shown by Planck's equation [38]:

$$E_\lambda = hc/\lambda \tag{1}$$

where $E_\lambda$ is the energy of a photon, $\lambda$ is the wavelength of the radiation, $h$ is the Planck's constant, and $c$ is the speed of light.

Different types of sources have been used in photochemical AOPs for dye degradation, such as high-pressure mercury lamps (infrared radiation, UV, and visible range), low-pressure mercury lamps ($\lambda_{max}$ = 254 nm), pulsed UV lamps, ultraviolet light-emitting diodes (UV LEDs), microwave-powered electrodeless discharge lamps ($\lambda_{max}$ = 254 nm), tungsten lamps ($\lambda > 410$ nm), and continuously operated xenon lamps ($\lambda$ = 400–1000 nm) [8]. Several factors can influence the efficacy of the radiation system, including wastewater characteristics, UV intensity, microbe exposure period to UV, and the reactor design [39].

Continuous-wave mercury vapor lamps are commonly used in photochemical AOPs, in which the lamps are arc discharge instruments that emit UV radiation by creating an electric arc between the two electrodes in specifically constructed lamps containing mercury vapor or a combination of mercury and another gas [40]. The energy released by the excitation of the mercury vapor results in the emission of UV radiation. However, there are concerns over mercury toxicity, high energy needs, mechanical instability, and the cleaning costs of the application of mercury lamps, thus emphasizing the need to find alternative UV sources [33].

## 5.1. Pulsed UV Lamps

Pulsed UV (PUV) lamps are a non-mercury alternative to mercury vapor lamps. PUV lamps are a method that was initially developed for and has been used extensively in the inactivation of bacteria and viruses and is now utilized in industry to decontaminate the cups used in food packaging [8,41]. PUV is based on the use of xenon lamps and it generates polychromatic high-intensity light in a very short period using high-power electrical pulses [33]. The release occurs in a non-toxic, rare-gas environment and can use flashlamps or surface discharge lamps. PUV lamps are instant-on lights that do not require the warm-up period which is required for mercury vapor lamps and also possess higher efficiency [33]. As PUV contains a significant amount of UV light, a potential application of this technology might be its usage as a part of an AOP for the breakdown of polluting dyes. A few studies to assess the performance of PUV have been conducted on dyes such as MG dye and azo dyes in the UV/$H_2O_2$ process [8,41].

## 5.2. Ultraviolet Light Emitting Diodes (LEDs)

A light-emitting diode (LED) is a semiconductor device that emits light in a restricted range using electroluminescence [33]. LEDs are made of non-toxic materials and use less energy than conventional lamps because they convert more energy into light and lose less energy as heat. Furthermore, LEDs are durable and emit only the required wavelengths that will save energy. The small size of an LED may be advantageous in some applications [33]. UV LEDs are considered an efficient technology for water treatment, as LEDs with low wavelengths have been developed.

## 5.3. Microwaved Electrodeless Discharge Lamp

Microwave-assisted electrodeless discharge lamps also have been widely used and can emit light covering a wide spectrum from vacuum UV to infrared when bombarded with an electromagnetic field [33]. Longer lives are predicted for these lamps since the light lacks an electrode [40]. The microwave power was totally used in the experimental setup to create UV radiation and volumetric heating, both of which contributed to the dye degradation process and resulted in no microwave power energy loss. MWEDL is considered to be effective on an industrial scale for water treatment as well, depending on the reactor configuration [33].

## 6. Photolysis

Photolysis is termed photodegradation by the irradiation of UV light applied to degrade the compounds that absorb light at shorter wavelengths [22]. The interaction of the light with molecules causes the molecules to degrade into simpler fragments [38]. This method's processes include UV/photolysis, UV/$H_2O_2$, UV/ozone, and photo-Fenton [22].

## 6.1. Direct Photolysis

UV/photolysis is a process that uses UV irradiation without oxidants and catalysts to break down the contaminants into smaller molecules [42]. The water molecule is cleaved under UV irradiation and transformed into hydrogen and hydroxyl radicals [20]. The photolysis reaction of dye can be represented by the following equation:

$$\text{Dye} + h\nu + (\text{oxygen}) \rightarrow \text{products} \tag{2}$$

$$H_2O + h\nu \rightarrow {}^{\bullet}OH + H \tag{3}$$

$$\text{Dye} + {}^{\bullet}OH \rightarrow \text{products} \tag{4}$$

### 6.2. Photolysis Method Based on $H_2O_2$ (UV/$H_2O_2$)

The UV/$H_2O_2$ process is a method that employs hydrogen peroxide in conjunction with UV radiation. Hydrogen peroxide involves external activation, such as UV light, and the photolysis of hydrogen peroxide will produce an effective oxidizing species, hydroxyl radical ($^\bullet$OH) [18]. The generation of hydroxyl radicals by the reaction of UV/$H_2O_2$ is as follows:

$$H_2O_2 + h\nu \rightarrow 2{}^\bullet OH \tag{5}$$

### 6.3. Photolysis Method Based on $O_3$ (UV/$O_3$)

In the UV/$O_3$ process, the generation of hydroxyl radicals occurs via ozone photolysis in the presence of water [18]:

$$O_3 + h\nu + H_2O \rightarrow 2{}^\bullet OH + O_2 \tag{6}$$

Table 3 shows previous studies on the degradation of dyes via photolysis.

**Table 3.** Various photolysis processes for degradation of dyes.

| Dye | Type of Photolysis | Experimental Conditions | COD/TOC/Degradation Percentage/other Remarks | Reaction Kinetics | Reference |
|---|---|---|---|---|---|
| MB—cationic dye, thiazine | UV/H$_2$O$_2$ | UV light source (UV lamp = 6 W) | - 99.86% of 10 ppm MB with 15 mL of 10% H$_2$O$_2$ after 60 min <br> - 99.22% of 20 ppm MB with 15 mL of 30% H$_2$O$_2$ for 60 min <br> - 98.90% of 30 ppm MB with 15 mL of 50% H$_2$O$_2$ after 90 min) | N/A | [39] |
| MB | UV | UV lamps (UV-A, UV-B, UV-C) | - COD removal: 83.3% achieved after 60 min (UV-C) <br> - 96.7% degradation of MB after 5 h (UV-C) | First-order | [42] |
| Blue 13 -monoazo dye | UV/H$_2$O$_2$ | UV-A (6, 12 and 18 W) | - 0.49–2.43% discoloration with 6 W light intensity <br> - 0.88–4.86% discoloration with 12 W light intensity <br> - 1.26–7.30% discoloration with 18 W light intensity | N/A | [43] |
| RhB | UV/H$_2$O$_2$ | [H$_2$O$_2$] = 0.10–1020 mg/L, pH 4.5, UV light (low pressure, $\lambda$ = 254 nm) | - 100% discoloration with 51 mg/L H$_2$O$_2$ in 6 min <br> - TOC: 69% reduced | N/A | [44] |
| RB 19—anionic, anthraquinone | UV, UV/O$_3$ | Two monochromatic germicidal lamps (40 W, 253.7 nm), T = 20 °C, [RB19]$_0$ = 230 $\pm$ 1.5 mg/L, pH 6.0, [O$_3$]$_0$ = 50 $\pm$ 2 mg/L | - Direct UV photolysis was negligible <br> - TOC: 94% decreased after 120 min | Pseudo-first-order | [28] |

**Table 3.** *Cont.*

| Dye | Type of Photolysis | Experimental Conditions | COD/TOC/Degradation Percentage/other Remarks | Reaction Kinetics | Reference |
|---|---|---|---|---|---|
| MO | UV | Medium pressure Hg lamp (150 W, 350–400 nm), pH 7.3, $[MO]_0 = 100$ µM | - 45.7% MO degraded after 120 min | Pseudo-first-order | [20] |
| Basic Red 1 (BR1)—cationic dye | UV/Fenton | Mercury lamp (9.5 W, 254 nm), pH 3.0, $[BR1]_0 = 100$ mg/L | - 90% degraded of BR1 after 60 min<br>- 70% of TOC is removed | N/A | [45] |
| Brilliant blue FCF (BBF) -Triarylmethane dye, anionic | UV/Cl | Radiation sources: UV lamp (4W, 254 nm, Philips) as UVC source, solar irradiation as an alternative | - Acidic pH is favorable<br>- Solar irradiation is an alternative to UVC lamp<br>- Degradation rate increases when iron and copper are present<br>- After 80 min, 37% of BBF was mineralized | Pseudo-first-order | [46] |
| Congo red (CR) -Azo dye | UV/NO$_3$ | Low-pressure UV (254 nm) | - The degradation rates were 81.90%, 17.46%, and 0.63% by OH$^\bullet$, NO$_2^\bullet$, UV radicals respectively<br>- Neutral conditions are favorable | Pseudo-first-order | [47] |
| MB—Cationic dye | VUV/UV/ Persulfate | $[MB]_0 = 10$ µM, $[PS]_0 = 0.5$ mM, T = 25 °C, Reaction time = 10 min. | - VUV generated more reactive OH$^\bullet$ and SO$_4^{\bullet-}$ radicals than a conventional UV/persulfate process.<br>- MB degradation increases with increasing pH solution (3.0–11.0) | N/A | [48] |

| Dye | Type of Photolysis | Experimental Conditions | COD/TOC/Degradation Percentage/other Remarks | Reaction Kinetics | Reference |
|---|---|---|---|---|---|
| Chlorazol black (CB) | UV/acetone | [CB] = 20 mg/L at 25 °C, [Acet.]$_0$ = 50 mM, pH = 3–9 | - CB was nearly full degraded after 30 min in the presence of acetone, which was 5.6 times greater compared to UV alone.<br>- Strong alkaline pH is favorable | First-order | [49] |
| Direct yellow 106 (DY106)—Azo dye, anionic | PL direct photolysis, PL/$H_2O_2$ | Pulsed light, [DY106]$_0$ = 20 mg/L, [$H_2O_2$]$_{0600}$ mg/L, pH 9.5 | - PL direct photolysis was negligible on the degradation due to the highly stable chemical structure of the dye.<br>- PL/$H_2O_2$ process reaches 90% of degradation. | Pseudo-first-order | [8] |
| C.I. Acid Blue 25 (AB25) Anthraquinone | Direct UV irradiation, UV/$H_2O_2$, UV/Fe(II) | Direct photolysis:<br>- effect of initial concentration: [AB25] = 10–150 mg/L, pH = 5.7, t = 20 min<br>- Effect of pH: [AB25] = 50 mg/L, pH = 1.4–5.7, t = 20 min | - 100% decolorization for 10 mg/L of AB25 with photolysis.<br>- Degradation rates in acidic solutions (1.4–3) are higher, decreasing from pH 1.4 to 5.7, and there is nearly no change in the pH range of 5.7–9.3. Higher decolorization rates are observed in basic media (10.5–11.8). | Pseudo-first-order | [50] |
| Reactive orange 16 (RO16)—anionic dye | UV, UV-C/$H_2O_2$ | UV-C germicidal tubes (8 W), pH = 6.5, t = 30 min | - Degradation by direct photolysis without $H_2O_2$ was negligible<br>- Decolorization rate increased with increasing in $H_2O_2$ | Pseudo-first-order | [51] |

**Table 3.** *Cont.*

| Dye | Type of Photolysis | Experimental Conditions | COD/TOC/Degradation Percentage/other Remarks | Reaction Kinetics | Reference |
|---|---|---|---|---|---|
| Reactive orange 16 (RO16)—anionic dye | UV, UV-C/$H_2O_2$ | UV-C germicidal tubes (8 W), pH = 6.5, t = 30 min | - Degradation by direct photolysis without $H_2O_2$ was negligible<br>- Decolorization rate increased with increasing in $H_2O_2$ | Pseudo-first-order | [51] |
| RO16 -Anionic mono-azo dye | UV/$H_2O_2$ | Low-pressure mercury vapor lamp (28 W, 253.7 nm) | - Neutral conditions are favorable<br>- Decolorization of 50.0 mg dm$^{-3}$ RO16 was completed in <6 min at pH 7, with a UV light intensity of 1950 µW cm$^{-2}$)<br>- Optimum $H_2O_2$ concentrations are in a range from 20 to 40 mM | Pseudo-first-order | [52] |
| RB19 | UV, UV/$H_2O_2$ | Low pressure mercury lamp (65 W, 254 nm), [RB19]$_0$ = 10–100 mg/L for photolysis, a fixed [RB19]$_0$ = 100 mg/L and [$H_2O_2$] = 100,300, 500 and 800 mg/L for UV/ $H_2O_2$, pH = 3 | - Dye degradation with UV photolysis is negligible as the DOC reduction was less than 2% after 300 min.<br>- 91% of RB19 was degraded after 3h of UV/$H_2O_2$ reaction at a concentration of 500 mg L$^{-1}$ $H_2O_2$ | N/A | [53] |
| Acid Orange 8 (AO8), Acid blue 29 (AB29), Acid blue 113 (AB113) (Azo dyes) | VUV/$H_2O_2$ | Low-pressure mercury 185 nm vacuum UV lamp (6 W), | - 90% of orange 8, 50% of acid Blue 29, and 60% of acid Blue 113 were degraded after 60 min of irradiation time. | First-order | [54] |
| Reactive red 120 (RR120) | UV/$Fe^3$ | Low pressure mercury lamp, [$Fe^{+3}$]$_0$ = 0.25–2.75 mM, [MB]$_0$ = 100–200 mg/L, initial pH 1–11 | - 92% color of the dye was removed in 55 min under conditions of $Fe^{3+}$ concentration, 2.35 mM, pH 3.6, and an initial dye concentration of 170 mg/L | N/A | [55] |

**Table 3.** *Cont.*

| Dye | Type of Photolysis | Experimental Conditions | COD/TOC/Degradation Percentage/other Remarks | Reaction Kinetics | Reference |
|---|---|---|---|---|---|
| Reactive green 19 (RG19) | UV/ $H_2O_2$ | Low-pressure mercury lamp (6 W, 254 nm), pH 2–10, $[RG19]_0 = 90$ mg/L | - Complete decolorization after 20 min<br>- 63% TOC removed in 90 min | Pseudo-first-order | [56] |
| Acid red 27 (AR27) (anionic dyes) | UV/$H_2O_2$ | Low-pressure mercury lamp (8 W) $H_2O_2$ (0.03% (*v*/*v*), $[RG19]_0 = 50$ µg mL$^{-1}$, sample flow rate of 6 mL min$^{-1}$ | - >99.9% degradation completed with hydroxyl radicals. | N/A | [27] |
| Basic Fuchsine dye | UV | UV-A light, pH 6.4 | - Acidic medium is favorable<br>- Degradation from 73.75% to 89.63% at 70 min | N/A | [57] |
| Brilliant green (BG) | UV | UV tubes (11 W, 350–450 nm), $[BG] = 10$–50 ppm | - Higher for the concentrations of 10 (61.5%) and 20 ppm (63.5%) and lower for initial concentrations of 30 (52.9%) and 40 ppm (40.7%). | First-order | [58] |
| Allura red | UV | T = 35 °C, t = 1–6 h, pH =3–12 | - After 1 h of irradiation, 95 % of Allura red (50 ppm) degraded at pH 12 and 35 °C | N/A | [59] |
| Erythrosine | UV | T = 35 °C, t = 1–6 h, pH =3–12 | - After 6 h of irradiation, 90% of Erythrosine (50 ppm) degraded at pH 6 and 30 °C | N/A | [59] |
| Remazol turquoise blue (RTB) | UV/$H_2O_2$ | UV lamp (6 W, 254 nm), $[RTB]_0 = 25$ ppm | - Under optimum conditions, degradation of the dye could achieve 50% in 10 min<br>- $PO_4^{3-}$; $Cl^-$, and $CO_3^{2-}$ increased the degradation rate of the dye | First-order | [60] |

| Dye | Type of Photolysis | Experimental Conditions | COD/TOC/Degradation Percentage/other Remarks | Reaction Kinetics | Reference |
|---|---|---|---|---|---|
| Acid red 94 (AR94)— xanthene dye | UV/$H_2O_2$ | UV lamp (254 nm), | - Maximum decolorization was 90% with conditions of 0.005 mM dye, at optimum 0.042 M $H_2O_2$ and pH 6.6. | Pseudo-first-order | [61] |
| Alizarin yellow (AY)—azo dye | UV/acetone, UV/$H_2O_2$, UV/$S_2O_8^{2-}$ | Low-pressure mercury lamps (15 W, 254 nm), T = 18 °C and 20 °C, pH = 1.7, 2, 11.5 and 12 | - The degradation rate of AY for a reaction time 40 min in ascending order: UV/acetone ($10^{-2}$ M, 17%) < UV/$H_2O_2$/$S_2O_8^{2-}$ ($10^{-2}$ M, 60%) < UV/ $S_2O_8^{2-}$ ($10^{-2}$ M, 81.37%) < $H_2O_2$/UV ($10^{-2}$ M, 97.68%) | N/A | [62] |
| Carmine (C.I. natural Red 4) | UV/$H_2O_2$ | UV lamp (254 nm), T = 25 °C, $[dye]_0$ = 20–160 μM, $[H_2O_2]_0$ = 0.83–6.64 mM, pH = 2–10, t = 30 min | - Optimum conditions of 62 μM dye are 5.5 mM $H_2O_2$ and pH 4. | N/A | [63] |
| Mordant red 73 (MR73) | UV/$H_2O_2$ | [MR73] = 0.1 mM, 0.05 mM, 0.05 mM, $[H_2O_2]_0$ = 2.5 mM, pH 3 and T = 25 °C | - MR73 decolorization was complete in less than 1 h under optimum conditions.<br>- 65% of MR73 was mineralized after 3 h. | Pseudo-first-order | [64] |
| Direct red 23 (DR23) AB25 Mordant Orange 1 (MO1) | UV/Fenton | Mercury lamp (9.5 W, 254 nm) | - 32% of TOC and 48% of the color of DR32 was removed after 1 h.<br>- Complete decolorization and more than 90% of TOC for AB25 and MO1. | First-order | [45] |
| Blue 13 Monoazo dye | UV/ $H_2O_2$ | UV lamp (6,12 and 18 W, 254 nm), Comparing the UV intensity | - 99.70% of dye decolorized at 0.67% $H_2O_2$ using 18 W intensity of UV light for 40 min at 7.86 pH. | N/A | [43] |

**Table 3.** *Cont.*

| Dye | Type of Photolysis | Experimental Conditions | COD/TOC/Degradation Percentage/other Remarks | | Reaction Kinetics | Reference |
|---|---|---|---|---|---|---|
| MG, Bromocresol purple (BCP) | UV | Low-pressure mercury lamp (15 W, 254 nm), T = 18–20 °C, pH 5.8 for MG and pH 4.5 for BCP | - | 25% of BCP degraded and 66% of MG degraded after 240 min. | Pseudo-first-order | [65] |
| Orange G Azo dye, anionic | UV, UV/acetone, UV/$H_2O_2$, UV/$S_2O_8^2$ | Low-pressure mercury lamp (15 W, 254 nm), T = 18–20 °C, pH 5.8 for MG and pH 4.5 for BCP | - - | The results obtained from direct photolysis of the dye were negligible. Complete decolorization by UV/acetone after 1 h. | N/A | [66] |
| RhB—xanthene | UV, UV $H_2O_2$, UV/Persulfate | Medium-pressure mercury lamp (330 W, 365 nm) | - - - | With only UV irradiation, the degradation of RhB reached 45% after 6 min. 96% and 87% of RhB were removed by the UV/$H_2O_2$ and UV/PS processes, respectively, after 15 min. TOC removal was 50% by UV/$H_2O_2$ process and 60% by UV/PS process. | Pseudo first order | [67] |
| Reactive black 5 (RB5)— anionic dye | UV, UV/$H_2O_2$ | Low-pressure mercury lamp (55 W), [RB5]$_0$ = 10–50 mg/L, t=120 min | - | 60% decolorization was achieved after 120 min at 50 mg/L of RB5 with UV irradiation. | Pseudo-first-order | [68] |
| Disperse orange 25—non-ionic | UV, UV/$H_2O_2$ | Low-pressure mercury lamp (55 W), [RB5]$_0$ = 10–50 mg/L, t = 120 min | - - | Low photodegradation rate because of low aqueous solubility (non-ionic dye). Only 10.6% decolorisation was achieved after 120 min | Pseudo-first-order | [68] |

| Dye | Type of Photolysis | Experimental Conditions | COD/TOC/Degradation Percentage/other Remarks | Reaction Kinetics | Reference |
|---|---|---|---|---|---|
| Basic blue 3 (BB3), Acid green 25 (AG25) | UV/ $H_2O_2$ | Mercury lamp (30 W, 254 nm), pH 6.5, $[H_2O_2]_0$ = 1.2 g/L, $[BB3]_0$ = 10 mg/L, $[AG25]_0$= 10 mg/L | - BB3 achieved 95.03% of removal after 20 min <br> - AG25 achieved 98.16% of removal after 20 min | Pseudo-first-order | [69] |
| MO | UV/ $H_2O_2$ | UV lamp (254 nm), $[MO]_0$ = $7.80 \times 10^{-5}$ M, $[H_2O_2]_0$ = $4.58 \times 10^{-2}$ M | - Complete degradation of 0.078 M of MO after 3 min and only 26% degradation of MO after 4 h without $H_2O_2$ | Pseudo-first-order | [70] |
| MB | UV/ $H_2O_2$ | Medium pressure lamp (300 W, 365 nm), | - Optimum pH and $H_2O_2$ dosage of the photolysis process were pH 4–5 and 0.165 mL 30% $H_2O_2$ per mg of MB. | First-order | [71] |
| Tartrazine | UV | Solar UV, UV lamp (24 W, 365 nm), t = 300 min, flow rate of solution = 60 mL/s, $[dye]_0$ = 10 mg/L, pH 8.2–8.5 | - 2% of dye was degraded after 300 min. | N/A | [72] |
| Indigo carmine | VUV | VUV light from Xe-Ne plasma (147–172 nm), | - No oxidizing agent or catalyst is required | N/A | [73] |

*6.4. Photo-Fenton*

Photo-Fenton is a process utilizing the combination of UV radiation with Fenton's reagents (solution of hydrogen peroxide with ferrous iron), and as the light irradiation increases, the Fenton process efficiency will also increase. Irradiation with UV light could speed up the degradation of organic pollutants. The reactions of photo-Fenton are shown in Equations (7)–(9). UV light causes not only the production of more hydroxyl radicals but also the reuse of ferrous catalysts through $Fe^{3+}$ reduction (Equation (8)) [45]. The decomposition of $Fe(OH)^{2+}$ also enhances more $^{\bullet}OH$ radicals (Equation (9)) [18]:

$$Fe^{2+} + H_2O_2 + h\nu \rightarrow {}^{\bullet}OH + Fe^{3+} + OH^- \tag{7}$$

$$Fe^{3+} + H_2O + h\nu \rightarrow {}^{\bullet}OH + Fe^{2+} + H^+ \tag{8}$$

$$Fe(OH)^{2+} + h\nu \rightarrow Fe^{2+} + {}^{\bullet}OH \tag{9}$$

## 7. Factors Affecting the Degradation Rate of Photolysis Methods

*7.1. Contact Time*

Previous studies have reported that dye concentration in aqueous solutions decreases with irradiation exposure time. Joseph et al. [42] have reported that MB concentration degraded more with longer irradiation time. Safni et al. [74] have also reported that the degradation percentage of orange F3R by photolysis increased with increasing irradiation time with or without catalysts. This is because the longer the UV irradiation time, the more energy is absorbed by dye molecules from photons which facilitate them to degrade. The degradation efficiency also increases with irradiation time, because more hydroxyl radicals can be generated to oxidize the organic pollutants to break down into simpler molecules [42,74].

*7.2. Radiation Source*

Primarily, artificial light sources are purposely used in the photolysis process in wastewater treatment to maintain stable intensities and avoid other environmental factors. The intensity and wavelength of the light source significantly affect the photodegradation rate [34]. A shorter wavelength gives higher-energy photons, which will increase the degradation efficiency [42]. The higher intensity means more photons can be absorbed to generate hydroxyl radicals. However, in most photolysis experiments, very high-intensity light is avoided due to the probability of temperature rise, which could cause a thermal reaction [34]. The effects of UV irradiation intensities have been studied by Jamal et al. [43] and Algubili et al. [75]. They reported that the degradation percentage of dyes increased after treatment with increasing intensities of UV irradiation. The effect of different types of UV irradiation has been proven by Joseph et al. [42], who found that between four types of irradiation, UV-C (254 nm), UV-B (311 nm), UV-A (365 nm), and a solar lamp (610 nm), UV-C degraded MB dye almost 80% in only 1 h compared to after 5 h using UV-B, UV-A, and solar, which were only 12.07, 38.24, and 29.56, respectively.

*7.3. pH of the Medium*

The pH of the medium plays a vital role as a parameter of the degradation efficiency of dyes due to the wide range of pH values of dye effluents. In previous studies, researchers have performed comparative experiments on different pH conditions in the photochemical process. Ghodbane and Hamdaoui [50] have reported that the degradation rate of AB25 in acidic solutions (1.4–3) is higher, decreases from pH 1.4 to 5.7, and there is nearly no change in the pH range of 5.7–9.3. Then, higher decolorization rates were observed in a basic solution (10.5–11.8) because of the higher facility of OH radicals generated at these pH values. Mullapudi et al. [27] have reported that the maximum degradation of AR27 and MB by $UV/H_2O_2$ was achieved in the pH range of 3–9. The efficiency of degradation was decreased when the pH of the solution increased from 10 to 11. It can be explained that at higher pH, $H_2O_2$ deprotonates to form hydroperoxy anion ($HO_2^-$) which is a conjugate

base of $H_2O_2$. The formed hydroperoxy anion reacts with $OH^-$ radicals and $H_2O_2$, yielding dioxygen and water, thereby reducing the availability of hydroxyl radicals to attack the target molecule and affecting the degradation efficiency [41].

$$OH^- + HO_2^- \rightarrow H_2O_2 + O_2^- \tag{10}$$

$$H_2O_2 + HO_2^- \rightarrow H_2O + O_2 + OH \tag{11}$$

*7.4. Initial Concentration of Dyes*

Many researchers have performed photolytic experiments by varying the initial concentration of dyes to investigate the effects on the degradation of dyes and organic pollutants. It is proven that the higher the initial concentration of dye, the lower the degradation rate. At high concentrations, the penetration of photons entering the solution diminishes, resulting in an inner filter effect and the solution is not accessible by UV light [75,76]. Bendjama et al. [49] have reported that the degradation efficiency of CB dye decreased with increasing initial concentration. For example, complete degradation was achieved for a dye concentration of 5 mg/L, and the result observed decreased to about 95% for 20 mg/L, 80% for 30 mg/L, and 37.4% for 50 mg/L of CB. Ghodbane and Hamdaoui [50] also reported that the degradation rate of AB25 dye decreased with an increasing initial concentration of dye. Similarly, it was also reported by Soltani and Entezari [77] that the photocatalytic degradation of MB dye is affected by the initial concentration, as the degradation rates were decreased when the initial concentration of MB was increased.

## 8. Photocatalysis

Photocatalysis is a heterogenous AOP using light irradiation and a semiconductor photocatalyst that absorbs light, which differs from other treatment techniques because this process performs oxidation and reduction simultaneously [22,78]. There are many semiconductor materials that have been used as catalysts for the degradation of organic pollutants and dyes, including $TiO_2$, ZnO, $Fe_2O_3$, $SnO_2$, $ZrO_2$, CdS, ZnS, and many more. A number of studies have demonstrated that photocatalysis is a reliable and efficient method for degrading dyes in water. Table 4 lists the photocatalytic dye degradation used in the previous literature.

**Table 4.** Various photocatalysts studied for the degradation of dyes.

| Dye | Photocatalyst | Experimental Conditions | COD/TOC/Degradation Percentage/Remarks | | Reaction Kinetics | Reference |
|---|---|---|---|---|---|---|
| MB | Ferrite Bismuth nanoparticles | Direct solar irradiation; $[MB]_0 = 15$ mg/L; acidic medium; photocatalyst (0.5 g/L) | - | 95% of MB was degraded after 80 min | Pseudo-first-order | [77] |
| RB5 | $TiO_2$ | Low-pressure Hg UV-C lamp (15 W, 254 nm); $[TiO_2]_0 = 0.5$ g/L; $[RB5]_0 = 25–125$ mg/L; pH 6.4–6.9 | - | 82% TOC and 76% COD removal were achieved after 210 min | Pseudo-first-order | [79] |
| Procion Blue HERD (PBH) | $TiO_2$ and ZnO | UV lamp (30 W); $[PBH]_0 = 10–100$ ppm; photocatalyst loading = 0.5–2 g/L; pH 2–10 | - | 100% decolorization of dye with ZnO at pH 7 and with $TiO_2$ at pH 4. | First-order | [80] |
| Methyl orange (MO) | Ag-doped titania-silica | Medium-pressure Hg lamp (150 W, 350–400 nm); $[MO]_0 = 100$ µM; pH 7.3; T = ~20 °C; 0.15 g catalyst | - | After 30 min, photolysis (45.7%) degraded more than photocatalysis. | Pseudo-first-order | [20] |
| Orange F3R | C-N-codoped $TiO_2$ | UV (10 W, 365 nm), visible (13 W), solar irradiation; $[dye]_0 = 30$ mg/L; dosage of C-N-codoped $TiO_2$ is 3–15 mg | -<br>-<br>-<br>- | 39.05% degraded under UV light after 180 min<br>40.86% degraded under visible light after 180 min<br>64.12% degraded under solar light with 6 mg catalyst after 180 min<br>11.82% without catalyst and the degradation rate increased to 39.05% with C-N-codoped $TiO_2$ | N/A | [74] |
| MG | $TiO_2$ | UV lamp (15 W, 365 nm); $[MG]_0 = 40$ mg/L; 20 mg of $TiO_2$ | - | 99.9% MG degraded after 1 h. | N/A | [81] |
| Remazol Brilliant Blue (RBB) | ZnO | High-pressure Hg lamp (125 W, 365 nm), $[ZnO] = 1.5$ g/L, $[RBB]_0 = 100$ mg/L | - | 100% degradation of RBB | Pseudo-first-order | [75] |

**Table 4.** *Cont.*

| Dye | Photocatalyst | Experimental Conditions | COD/TOC/Degradation Percentage/Remarks | Reaction Kinetics | Reference |
|---|---|---|---|---|---|
| MB | ZnO/PDMS | Three different types of light sources such as halogen (100 W), metal-halide (150 W), and UV (4 W) light sources. | The highest degradation of MB achieved was 93% under UV/Vis irradiation after 3 h. | N/A | [26] |
| Rhodamine B (RhB) | $TiO_2$ | Low-pressure UV lamp (15 W, 254 nm), 120 mg $TiO_2$, pH 4.5, $[RhB]_0 = 5$ mg/L | Color removal achieved 29% after 60 min and TOC, 25% | Pseudo-first-order | [44] |
| RO16—anionic monoazo dye | $UV-C/TiO_2$ and $UV-C/H_2O_2/TiO_2$ | UV-C germicidal tubes (8 W), pH 6.5 | - $UV-C/H_2O_2$ is the most effective process compared to $UV/TiO_2$ or $UV/H_2O_2/TiO_2$. | Pseudo-first-order | [51] |
| MO, RhB, MB | Ag-doped titania-silica | Medium-pressure mercury lamp (150 W, 350–400 nm), Catalyst was prepared by sol-gel, catalyst loading is 0.15 g, T = 20 °C | - The degradation rate of MO by photolysis was higher than photocatalysis<br>- The degradation of (RhB) by photocatalysis was higher than photolysis, after 30 min | Pseudo-first-order | [20] |
| Violet-3B | C-N-codoped $TiO_2$ | Visible-halogen lamp (500 W), $[dye]_0 = 5$ mg/L, catalyst dosage is 0.3 g/L, pH 5.6 | - The degradation of violet-3B with visible light and catalyst after 4 h achieved 96% color removal and, due to mineralization, 44% TOC was removed. | Pseudo-first-order | [82] |
| Tartrazine | $UV/TiO_2$, $UV/H_2O_2/TiO_2$ | UV lamp (6 W, 254 nm), $[dye]_0 = 2 \times 10^{-5}$ to $8 \times 10^{-5}$ M, pH 2.2–11, 0.02–0.18 mg/L catalyst dosage, T = 30 °C | - The optimum conditions for the degradation of dye were $6 \times 10^{-5}$ M dye concentration, pH of 11, and 0.18 mg/L of catalyst dose.<br>- The most effective degradation of the dye was achieved with the combination of $UV/H_2O_2/TiO_2$ | Pseudo-first-order | [83] |

| Dye | Photocatalyst | Experimental Conditions | COD/TOC/Degradation Percentage/Remarks | Reaction Kinetics | Reference |
|---|---|---|---|---|---|
| MG | ZnO–TiO$_2$/clay | UV-A lamp (100 W, 365 nm), catalyst dosage = 1 g, [dye]$_0$ = 75 mg/L, pH 5.2 | - Nearly complete mineralization was achieved after 30 min. | Pseudo-first-order | [84] |
| MG | TiO$_2$ dip-coating | UV lamp, solar irradiation, | - Complete removal in the presence of catalyst after 6 h of sunlight irradiation <br> - 92.15%, 94.28% and 98.43 % for 5, 10 and 15 g of catalysts respectively | Pseudo-first-order | [85] |
| AO8, AC29, AB113 (Azo dyes) | VUV/TiO$_2$ | Low-pressure Hg lamp (18 W, 185 nm), TiO$_2$ dosage = 0.5 g/L, [dye]$_0$ = 0.0523 mM, T = 25 °C, pH 3, 5, 7, 9 and 11 | - Degradation of the dyes is more efficient under an acidic medium | Pseudo-first-order | [54] |
| MB | TiO$_2$, Sn–F/TiO$_2$ NPs | Sol-gel method, UV, and visible light irradiation | - Under UV irradiation, the degradation efficiency of MB was 72% and 91% for TiO$_2$ and Sn-F/TiO$_2$ NPs, respectively <br> - Under visible light, the degradation efficiency was 91.6% and 94.4% for TiO$_2$ and Sn-F/TiO$_2$ NPs, respectively | Pseudo-first-order | [86] |
| RB5—anionic dye | TiO$_2$ | UV lamps (40 W, 365 nm), pH (3–11), catalyst load (0.5–3.0 g/L), and [RB5]$_0$ = 20–100 mg/L | - Maximum degradation rate of 26.5 mg/g of dye after 30 min was at pH 3 and a catalyst load of 1.5 g/L <br> - TOC reduced by 70% | Pseudo-first-order | [87] |
| MB—cationic dye | TiO$_2$, TiO$_2$ ENR | Fluorescent lamp, | - Alkaline medium was favourable | Pseudo-first-order | [88] |

**Table 4.** *Cont.*

| Dye | Photocatalyst | Experimental Conditions | COD/TOC/Degradation Percentage/Remarks | Reaction Kinetics | Reference |
|---|---|---|---|---|---|
| MB | $TiO_2$ | UV-A, UV-B, UV-C and solar light, $[MB]_0 = 2–10$ ppm, pH (4–10), t = 1 h. | - Complete degradation of MB was achieved within 14 min with UV-C irradiation, 18 min with UV-B irradiation, and 20 min with UV-A irradiation | First-order | [89] |
| RB5—azo dye | ZnO, $TiO_2$ | Catalyst load = 0.5–1.5 g/L, $[RB5]_0 = 25–150$ mg/L), pH = 3.0–11.0 | - The color removal efficiencies of RB5 were 99.8% and 58.1% using ZnO and $TiO_2$ after 60 min.<br>- Rapid color removal with pH between 3 and 9 | Pseudo-first-order | [90] |
| RB5 | $TiO_2$, ZnO | UV lamp (20 W, 365 nm), Catalyst load = 1.25 g/L, $[RB5]_0 = 10–100$ mg/L), pH = 3.0–11.0 | - Complete decolorization with ZnO and 75% decolorization with $TiO_2$ after 7 min<br>- Acidic medium is preferred. | Pseudo-first-order | [91] |
| Reactive orange 4 (RO4) | $TiO_2$, ZnO | UV lamp (20 W, 365 nm), Catalyst load = 1.0 g/L, $[RO4]_0 = 10–100$ mg/L), pH = 3.0–11.0 | - 92% decolourization with ZnO and 62% decolorization with $TiO_2$ after 7 min | Pseudo-first-order | [91] |
| MB—cationic | Ta-doped ZnO | Xe arc lamp (300 W), $[MB]_0 = 10$ mg/L, 50 mg, pH 8 | −1 mol% Ta-doped ZnO annealed at 700 °C exhibits the highest degradation rate. | Pseudo-first-order | [92] |
| AO7 | $TiO_2$ | High-pressure mercury lamp (400 W) | - Complete removal of 0.086 mM AO7 after 20 min at pH 6.8 | First-order | [93] |
| Reactive red | $TiO_2$ | High-pressure mercury lamp (400 W) | - Complete removal of 0.086 mM of RR2 after 20 min at pH 6.8 | First-order | [93] |

**Table 4.** *Cont.*

| Dye | Photocatalyst | Experimental Conditions | COD/TOC/Degradation Percentage/Remarks | Reaction Kinetics | Reference |
|---|---|---|---|---|---|
| Reactive red | $TiO_2$ | High-pressure mercury lamp (400 W) | - Complete removal of 0.086 mM of RR2 after 20 min at pH 6.8 | First-order | [93] |
| Procion yellow H-EXL | N-doped $TiO_2$ | UV lamp (100 W) | - Optimum conditions were found to be at pH 5 with a $TiO_2$ dosage of 1 g/L. | N/A | [94] |
| Tartrazine—anionic azo dye | $TiO_2$ | Solar UV, UV lamp (24 W, 365 nm), t = 300 min, flow rate of solution = 60 mL/s, $[dye]_0$ = 10 mg/L, $TiO_2$ dosage = 0.3 mg/cm$^2$, pH 8.2–8.5 | - The removal efficiency was 97% for the solar reactor and 30% for the lamp reactor.<br>- Acidic medium is preferred. | N/A | [72] |
| CV, Methyl red (MR), Basic blue (BB) | ZnO, $TiO_2$, SnO | Solar irradiation, $[dye]_0$ = 10 mg/L, pH 9 | - COD removal: 92% for CV, 95% for BB, and 89% for MR after 5 h irradiation with the presence of ZnO. | N/A | [95] |
| MG | ZnS, Mn-doped ZnS | Medium-pressure lamp (125 W), pH 2–5, t = 90 min, $[MG]_0$ = 25 g/L | - Degradation efficiency increased from 60 to 72% as the pH increased from 2 to 4 | Pseudo-second-order | [96] |
| Reactive red 4 (RR4) | $TiO_2$, N-doped $TiO_2$ | LED light irradiation, $[RR4]_4$ = 30 mg/L, 0.030 g catalyst | - Complete degradation of RR4 after 60 min with N-doped $TiO_2$<br>- 40% of RR4 was removed by undoped $TiO_2$ after 30 min and no change thereafter. | N/A | [97] |
| MB | SnO | Low-pressure mercury lamp (125 W, 254 nm), 0.02 g $SnO_2$, $[MB]_0$ = 10 mg/L, | - No degradation by visible light.<br>- Complete degradation with UV irradiation after 30 min with SnO. | First-order | [98] |

| Dye | Photocatalyst | Experimental Conditions | COD/TOC/Degradation Percentage/Remarks | Reaction Kinetics | Reference |
|---|---|---|---|---|---|
| MB, RB | Fe-doped NiO | Sunlight radiations, $[RB]_0$ = 5 ppm, $[MB]_0$ = 5 ppm, t = 60 min, | - 86% of RB was degraded and 85% of MB was degraded after 60 min. | Pseudo-first-order | [99] |
| MB | Ni-doped $ZrO_2$ | Visible light lamp (>400 nm), $[dye]_0$ = 5 ppm, 15 mg of photocatalyst | - 92.2% degradation of dye after 100 min under visible light irradiation. | Pseudo-first-order | [100] |
| MB, MO | Pd-doped $TiO_2$ | High-pressure lamp (100 W), $[MB]_0$ = 20 mg/L, $[MO]_0$ = 20 mg/L, | - Without catalyst, 15.6% of MB and 87.8% of MO degraded under irradiation after 120 min.<br>- Complete decolorization for both MB and MO after 120 min. | Pseudo-first-order | [101] |
| MB | Co-doped $TiO_2$ | UV-C lamp, $[MB]_0$ = 10 ppm, 0.5 g/L catalyst | - Without a catalyst, 1.3% of MB was degraded after 100 min.<br>- 81.4% degradation of MB with catalyst after 100 min | Pseudo-first-order | [102] |
| RB | ZnO nanoparticles | UV lamp (15 W, 256 nm), 0.05 g ZnONPs, $[dye]_0$ = 20 mg/L, pH 4,8 and 11 | 96, 100 and 83% of RB degraded at pH 4, 8 and 11 respectively after 240 UV irradiation | First-order | [103] |
| MB | Immobilised $TiO_2$ | $[MB]_0$ = 75 mg/l, [Zinc] = 60 mg/L, [NaCl] = 0.250 M, flowrate of 0.7 L/min. | After 180 min of UV radiation, a 79.27% reduction in initial dye concentration was observed. | N/A | [104] |

### 8.1. Photocatalysts

8.1.1. Titanium Dioxide, $TiO_2$, and Metal Oxide Semiconductors

Among the various photocatalysts, $TiO_2$ is practically the only material appropriate for industrial applications at the moment and, most likely, in the future. This is due to the fact that $TiO_2$ has the most efficient photoactivity, the best stability, and is inexpensive and readily available [44,78]. $TiO_2$ degrades well-known cationic dyes such as MB, MG, MR, and basic violet. Basic dyes with high molecular weight are vulnerable to $OH^\bullet$ and radicals produced by $TiO_2$ [105]. Other than $TiO_2$, metal oxide semiconductors such as $SnO_2$, ZnO, NiO, $ZrO_2$, and ZnS have been used as photocatalytic materials. Among the metal oxides, ZnO has also been utilized as a photocatalytic material in wastewater treatment for decolorization It is low-cost, chemically stable, and has a larger light absorption spectrum than $TiO_2$ [105].

8.1.2. Modifications to Enhance Photocatalyst Activity

Metal Doping

The drawback of using $TiO_2$ as a photocatalyst is that it can be operated only under UV irradiation (250 nm–350 nm) [105]. Because of the magnitude of its band gap energy (3.0–3.3 eV), $TiO_2$ absorbs only a small percentage of the solar spectrum radiation [106,107]. In order to enhance the photocatalytic activity of $TiO_2$, the researchers make nanocomposites with metal and non-metal because it can be enhanced by decreasing the unwanted recombination of photoinduced holes and electrons and expanding the catalyst's photo response to the visible light area [108]. Metal ions such as manganese (Mn), zinc (Zn), aluminum (Al), iron (Fe), nickel (Ni), silver (Ag), palladium (Pd), chromium (Cr), vanadium (V), and platinum (Pt) have been widely used for doping with catalysts in order to minimize the band gap energy, which will improve the efficiency of UV light photocatalysis [108]. However, metal doping was also reported to have thermal instability and an increase in carrier trapping, which may decrease the photocatalytic efficiency [106]. Tolia et al. [96] have investigated the photocatalytic degradation of MG dye using Mn-doped ZnS and undoped ZnS nanoparticles. They found that with the doped photocatalyst, the rate of decolorization was increased as the Mn ion also functions as an oxidizing agent. However, it was discovered that increasing the doping concentration decreases photodegradation efficiency because the increase in manganese ions replaces Zn ions in the ZnS lattice structure and creates MnS, which reduces the activity of trapping electrons or holes.

Non-Metal Doping

Doping with non-metal also can enhance the efficiency of photocatalysis by suppressing electron-hole recombination and increasing the redox potential of $OH^\bullet$. Non-metal materials that have been utilized for doping are S, C, B, N, P, I, and F [109]. The inclusion of a co-dopant in photocatalysis can result in fast recombination of hole-electron pairs, implying that co-dopant $TiO_2$ has greater photocatalytic degradation efficiency than single dopant $TiO_2$. However, too much $TiO_2$ doping would be unfavorable to the photocatalytic activity of the photocatalyst. Thus, an optimal amount of dopant is utilized to optimize the effectiveness of the photocatalytic activity [108]. Nitrogen doping onto $TiO_2$ is considered due to high stability, small ionization, and comparable atomic size with oxygen that is better than standard $TiO_2$ and shows improved photocatalytic activity in the visible region [107]. In the photocatalysis of reactive red 4 (RR4), complete degradation was observed with N-doped $TiO_2$ after 60 min of LED light irradiation compared to only 40% or RR4 dye removed by undoped $TiO_2$ [97]. The result clearly demonstrated that the N element in $TiO_2$ lowered the band-gap energy, resulting in the photocatalyst being active under low visible light energy.

### 8.2. Mechanism of Photocatalysis

The schematic diagram of the general mechanism of photocatalysis is illustrated in Figure 2.

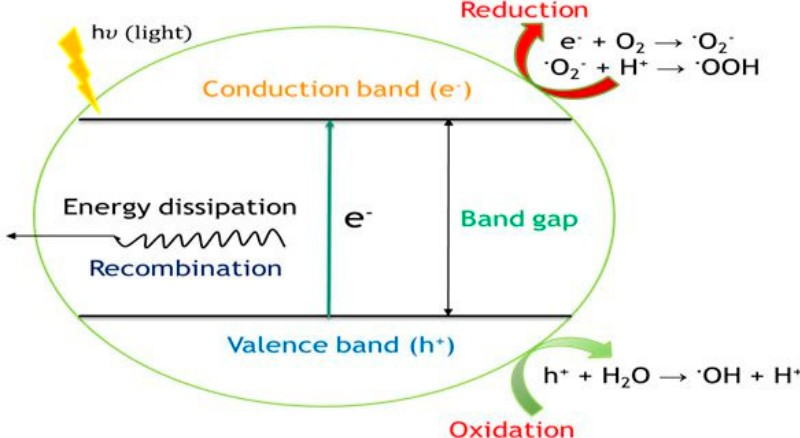

**Figure 2.** Schematic diagram of the mechanism of photocatalysis. Source: Mishra et al. [78].

The transport of pollutants from the surrounding environment to the photocatalyst surface is the first step in photocatalysis. The dye compounds are initially adsorbed onto the surface, where oxidation and reduction reactions take place [105]. The photocatalyst usually has band structures that include an empty conduction band and a filled valence band. When light strikes a semiconductor, its energy photons exceed the semiconductor's energy gap, and electrons are excited from the valence band to the conduction band by leaving holes behind (Equation (12)) [78]. The activated holes in the valence band ($h_{VB}^+$) react with water, generating hydroxyl radicals ($^\bullet OH$) (Equation (13)) and electrons in the conduction band ($e_{CB}^-$), generating superoxide radicals (Equation (14)) [78]. These radicals then degrade the pollutants into intermediates and further degrade to obtain $H_2O$ and $CO_2$ (Equations (15) and (16)) [22]. The products are then desorbed from the surface of the photocatalyst and moved back into the aqueous phase [105]. The summarized reactions in the photocatalysis process are as follows:

$$\text{Photocatalyst} + h\nu \rightarrow h_{VB}^+ + e_{CB}^- \tag{12}$$

$$H_2O + h_{VB}^+ \rightarrow {}^\bullet OH \text{ (hydroxyl radical)} + H^+ \tag{13}$$

$$O_2 + e_{CB}^- \rightarrow O_2 \text{ (superoxide radical)} \tag{14}$$

$${}^\bullet OH + \text{Pollutant} \rightarrow \text{Intermediates} \rightarrow H_2O + CO_2 \tag{15}$$

$${}^\bullet O_2 + \text{Pollutant} \rightarrow \text{Intermediates} \rightarrow H_2O + CO_2 \tag{16}$$

*8.3. Factors Affecting Photocatalysis Process on Degradation of Dyes*
8.3.1. Effect of pH

The key mechanism of the adsorptive removal of dyes in an aqueous solution is electrostatic interactions between dyes and metal catalysts. Metal hydroxyl groups (M-OH) are formed as a result of the adsorption of $H_2O$ molecules and the dissociation of $OH^-$ groups at the metal surface. When the solution pH is lower than the zero-point charge (ZPC) of the photocatalysts, the surface is positively charged (Equation (16)). On the other hand, when the solution pH is greater than the ZPC of the photocatalysts, the surface is negatively charged (Equation (17)) [5,88,89]. A high adsorption capacity is observed when anionic or cationic dyes are adsorbed on the photocatalyst surface at acidic or basic pH [5].

$$\text{when pH} < \text{pH}_{zpc} \quad \text{M-OH} + H^+ \leftrightarrow \text{M-OH-H}^+ \tag{17}$$

$$\text{when pH} > \text{pHzpc} \quad \text{M-OH} + OH^- \leftrightarrow \text{M-O}^- + H^+ \tag{18}$$

Laohaprapanon et al. [90] reported that the rapid color removal rates of RB5 under pH 3–9 were related to the photocatalytic activity of ZnO ($\text{pH}_{zpc}$ = pH 9). This can be

explained on the basis of the electrostatic interactions between the anionic dyes with the positive catalyst surface of ZnO under lower pH. However, they also reported that the highest color removal rate was found at pH 11 due to direct dye photolysis and the photocatalytic activity of ZnO. Although lower adsorption at high pH, the possible reason is due to the fact that more hydroxyl radicals are generated from a high concentration of hydroxyl ions. A similar result has been reported by Kansal et al. [91] for RB5 and RO4. Juang et al. [93] also reported that an acidic medium is favorable for the photodegradation process of AO7 and RR2. Jawad et al. [88] reported that MB, which is basically a cationic dye, has the highest photocatalytic efficiency by $TiO_2$ ($pH_{zpc}$ = pH 6.8) in an alkaline medium. This is because, at high pH, the $TiO_2$ surface is negatively charged, which causes more adsorption on the catalyst surface due to electrostatic interaction, and also more hydroxyl radicals are generated because more hydroxyl ions are available on the surface.

$$TiOH \leftrightarrow TiO^- + H^+ \tag{19}$$

### 8.3.2. Effect of Photocatalyst Loading

The photodegradation of dye increases as the amount of the photocatalyst increases. With increasing catalyst concentration, the number of active sites on the surface of the photocatalyst also increases. As a result, the production of OH radicals is increased, which is responsible for the degradation of dye solution [35]. Putri et al. [82] investigated the effect of photocatalyst—C-N-codoped $TiO_2$—dosage on the degradation of violet-3B. They found that the removal percentage of the dye was increased up to 83% as the catalyst dosage increased up to 0.3 g/L. However, the removal percentage was decreased when the dosage was more than 0.3 g/L, which is due to the increased turbidity of the dye solution which results in the lower penetration of light and increased light scattering [105]. Similar results were reported by Gupta et al. [83] in photocatalysis of tartrazine dye using $TiO_2$, wherein, by increasing the catalyst dosage from 0.02 g/l to 0.18 g/L, the degradation rate was increased, and then, with a catalyst dose of more than 0.18 g/L, the degradation remained constant. They also explained that when the amount of catalyst increases, the total active surface also increases. Simultaneously, UV-light penetration and the photoactivated volume of the suspension will be reduced due to the high dosage of the photocatalyst.

### 9. Energy Consumption and Cost–Benefit Analysis

UV-based processes for water treatment depend on the energy consumption in which electrical energy consumption represents a large amount of energy used. As a result, the decreased energy consumption would result in a larger yield and more accessibility of the approach. The amount of electrical energy consumed is determined by a variety of parameters, including the light source, the structure of the pollutants, and the type of reactor used [110]. Since electrical energy is so crucial in AOPs, the IUPAC has proposed a standard for addressing electrical energy used and reaction yield in UV-based AOPs [111].

$$Electrical\ energy\ consumption,\ E_{EO}\ (kWh) = \frac{1000 \times P \times t}{60 \times V \times log\frac{C_0}{C_t}} \tag{20}$$

where $P$ is the power of the used light source ($kW$), $t$ is irradiation time (min), $V$ is the volume of experimented solution (l), $C_0$ and $C_t$ are the initial and final concentrations of contaminants. $E_{EO}$ is the amount of energy required to degrade the pollutant's concentration in 1 m$^3$ volume ($kWh$).

Mohajerani et al. [13] stated that $UV/H_2O_2$ operation normally has a fixed cost of $58,000 and the cost of the UV lamp is $15,000 per year for a treatment plant. T The reduction in the number of UV lamps and reactors needed for the effective treatment of wastewater by the presence of $H_2O_2$ and increasing the dose of $H_2O_2$. An economic analysis of photocatalysis has been studied by Raju et al. [112], by comparing the cost of the process of UV alone, UV-$TiO_2$, granular activated carbon (GAC) alone, UV-GAC, and UV-GAC-$TiO_2$. The analysis of the cost is in terms of US Dollars (USD) per kilogram (kg)

of total volatile solids (TVS) utilized by the experimental results using the Equations (20) and (21).

$$Total\ power\ consumed,\ (kWh) = \frac{Power\ used\ (W) \times Reaction\ time\ (min)}{(1000 \times 60)} \tag{21}$$

$$\begin{aligned} Total\ operation\ cost\ &\left(\frac{USD}{kg}\right) \\ &= \frac{Total\ power\ consumed\ (kWh) \times Unit\ cost\ of\ power\ \left(\frac{USD}{kWh}\right)}{(C_0 - C_F)\left(\frac{mg}{l}\right) \times Working\ volume\ (l)} \times 10^6 \left(\frac{mg}{kg}\right) \end{aligned} \tag{22}$$

where $C_0$ is the initial concentration of TVS, $C_F$ is the final concentration of TVS, and the total power consumed involves the power consumed for the peristaltic pump and/or UV lamps and operating the stirrer. Among the processes, they concluded that UV-GAC-TiO$_2$ photocatalysis is the most cost-effective for wastewater treatment under batch and continuous operations.

## 10. Reaction Kinetics Model

In the process of photolysis, kinetic studies indicate the information on the optimum conditions for the degradation of dyes such as the initial concentration of dyes. The degradation kinetics of dyes can be determined by regressing the data obtained for the first-order kinetic Equation (22), pseudo-first-order kinetic Equation (23), second-order kinetic Equation (24), and pseudo-second-order kinetic equation Equation (25) [113]. The kinetic orders are being calculated to determine which order will fit the experimental data the best.

$$ln\ A_t = ln\ A_0 - k_1 t \tag{23}$$

$$ln\ (A_t - A_e) = log\ (A_t - A_e) - k_1(2.303)t \tag{24}$$

$$1/A_t = 1/A_0 + k_2 t \tag{25}$$

$$t/A_t = 1/(k_2 A e^2) + (1/Ae)t \tag{26}$$

where $A_0$ is the initial absorbance of the dye (A), $A_t$ is the effluent absorbance of the dye at time t, $A_e$ is the equilibrium absorbance of the dye, and $k_1$ (min$^{-1}$) and $k_2$ (A$^{-1}$ min$^{-1}$) are the rate constants of the first-order and second-order kinetic equations [113]. Multiple studies of the photodegradation of dyes have been proven to be fitted to first-order reactions and pseudo-first-order reactions.

A mathematical model was studied under the heterogeneous photocatalytic degradation of wastewater containing petroleum [114]. The various parameters influencing the mechanism of degradation, such as the mass transfer step, the kinetics of the mineralization, etc., were considered as the model parameters. Mass balances were chosen as the bulk region and the catalyst phase was considered as the solid phase in order to develop the model. The degradation mechanism of the solid phase was considered in two stages, such as an equivalent intermediate (EI), in which the toluene is transformed to EI, and in the second stage, the oxidation of EI gives carbon dioxide (CO$_2$). The results found a good correlation between modeling and empirical data in terms of degradation and mineralization. The simulation of the degradation kinetics of the target pollutant was obtained in the absence of a reaction pathway. A good similarity with the experiment was obtained for the target pollutant from the model.

## 11. Conclusions

One of the reasons for water pollution is the discharge of dye effluent into water bodies. Dye effluents in wastewater must be treated using efficient dye removal processes before being released into the environment, as they are one of the causes of water pollution. It was confirmed that AOPs that facilitate the combination of ultraviolet irradiation, catalysts, and oxidants to produce hydroxyl radicals ($^\bullet$OH) in solutions can be used as an effective technology for the degradation of dye-containing wastewater.

- AOPs are confirmed as a highly competitive technology in water treatment for removing organic pollutants, especially dyes.
- Different types of AOPs, such as photocatalysis, photolysis, $UV/H_2O_2$, photolysis, $UV/O_3$, photo-Fenton, electrochemical oxidation, ozonation, sonolysis, etc., can be effectively used for the treatment of dye-containing wastewater.
- This paper thoroughly investigated the current utilization of photolysis and photocatalytic treatment processes, which are among effective AOPs, for the degradation of dye-containing wastewater.
- The study confirmed that photocatalysis could be used for the complete mineralization of various dyes present in water using light and a photocatalyst by the simultaneous occurrence of oxidation and reduction reactions.
- pH, the initial concentration of the dye, catalyst loading, etc., were identified to have an influence on the photocatalytic degradation of the dye.
- A recent development in the photocatalytic process using $TiO_2$ is the photocatalyst modification by metal and non-metal doping, which results in improved photocatalytic activity in the presence of visible radiation.
- Utilizing a cost-effective and sustainable energy source for the photocatalytic degradation of pollutants is very effective.
- Sunlight as energy for the photocatalytic degradation of various dyes and other organic pollutants will be more efficient in terms of energy utilization.
- The combination of various AOPs such as photocatalysis combined with sonolysis, ozonation, electrolysis, Fenton, etc., will be major aspects for the complete mineralization of various organic pollutants. It will be effective, as there is a synergistic effect, by combining one or more AOPs, which will eliminate the drawbacks of individual processes.
- Scaling up an energy-efficient, cost-effective, and sustainable technique for the complete mineralization of various types of organic pollutants from water by using a reusable energy source, sunlight, and individual AOPs or hybrid AOPs is the challenging future aspect.

**Author Contributions:** S.M.A. Writing—review & editing. C.G.J. Supervision, Funding acquisition, Resource, Project administration, Writing—review & editing. C.K.P. Writing—review & editing. N.A.A. Writing—review & editing. S.N.M. Investigation, Writing—original draft. V.V. Writing—review & editing. All authors have read and agreed to the published version of the manuscript.

**Funding:** This research was funded by Research Management Center of Universiti Malaysia Sabah grant number GUG0315-1/2019. And The APC was funded by Research Management Center of Universiti Malaysia Sabah.

**Data Availability Statement:** Not applicable.

**Acknowledgments:** This research was supported by the Research Management Center of Universiti Malaysia Sabah (Grant No. GUG0315-1/2019). These contributions are gratefully acknowledged.

**Conflicts of Interest:** The authors declare no conflict of interest.

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
