# Peer review of "Current Trends in the Utilization of Photolysis and Photocatalysis Treatment Processes for the Remediation of Dye Wastewater: A Short Review"

_2305-7084, doi:10.3390/chemengineering6040058_

Round 1

Reviewer 1 Report

Main question addressed by the research: The work addresses a review of current trends on the utilization of photolysis and photocatalysis treatment processes for the remediation of dye wastewater.

Originality and relevance of the topic: The topic is relevant to the field and it considers a suitable review for the topic.
Added value of the paper:  The review takes into account the study of the different alternatives in terms of process enhancement and materials, however the main purpose of it is not clearly stated. The justification for this review should be included what aspects are critical for these assessments and clearly explain why they are analysing those and why they are needed at the end of the Introduction.

Specific improvements for the paper to be considered:

  1. Some aspects related to kinetics model in section 10 should be expanded as only two references are discussed.
  2. The conclusions are poor and they would need more elaboration so they clearly match the main findings. Conclusions should be divided in bullet points to discuss the main outcomes from important sections.

Author Response

Reviewer # 01 _ comments and our responses

Manuscript ID: ChemEngineering-1785338

Title: Current trends on the utilization of photolysis and photocatalysis treatment processes for the remediation of dye wastewater: A short review

General comments:

Main question addressed by the research: The work addresses a review of current trends on the utilization of photolysis and photocatalysis treatment processes for the remediation of dye wastewater.

Originality and relevance of the topic: The topic is relevant to the field and it considers a suitable review for the topic.

Added value of the paper:  The review takes into account the study of the different alternatives in terms of process enhancement and materials, however the main purpose of it is not clearly stated.

The justification for this review should be included what aspects are critical for these assessments and clearly explain why they are analysing those and why they are needed at the end of the Introduction.

Our responses:

The justification for this review is included and added the text at the end of Introduction section:

Ensuring the availability of clean water is essential for humans, terrestrial and aquatic animals and plants. In compliance to the sustainable development goals (SDG) to ensure this availability in the pursuit of global economic growth and industrial developments, generated wastewater has to be treated and remediated for reuse. Textile industries commence one of the most labour intensive industries providing employment to various downstream and upstream sectors. At the same time, this industry produces a large amount of dye contaminated wastewater that is release into rivers and streams. The modern dyes are synthetically designed to withstand weathering processes or biodegradation. Over the last few decades, researchers and scientists have investigated new techniques and methods to treat and remediate dye contaminated wastewater. This review paper explores the current and updated trends in this field with the focus on photolysis and photocatalysis treatment process.  

  • Some aspects related to kinetics model in section 10 should beexpanded as only two references are discussed.

Our responses:

Expalined and added into the text as,

“A mathematical model was studied under the heterogeneous photocatalytic degradation of wastewater containing petroleum [115]. The various parameters influencing the mechanism of degradation, such as mass transfer step, kinetics of the mineralization etc., were considered as the model parameters. Mass balances were chosen as bulk region and the catalyst phase was considered as the solid phase in order to develop the model. The degradation mechanism of solid phase was considered in two stages such as an equivalent intermediate (EI), in which the toluene is transformed to EI and in second stage, oxidation of EI gives carbon dioxide (CO2). The results found a good correlation between modeling and empirical data in terms of degradation and mineralization. The simulation of degradation kinetics of the target pollutant was obtained in the absence of reaction pathway. A good similarity with the experimental was obtained for the target pollutant from the model.”

  • The conclusions are poor and they would need moreelaboration so they clearly match the main findings.Conclusions should be divided in bullet points to discuss themain outcomes from important sections.

Our responses:

The conclusions are thoroughly edited and revised. Conclusions are divided in bullet points as suggested.

One of the reasons of water pollution is the discharge of dye effluent into water bodies. Dye effluents in wastewater must be treated using efficient dye removal processes before being released into the environment as it is one of the causes of water pollution. It was confirmed that AOPs which facilitate the combinations of ultraviolet irradiation, catalysts and oxidants to produce hydroxyl radicals (OH) in solutions can be used as an effective technology for the degradation of dye contained wastewater.

  • AOPs are confirmed as a highly competitive technology in water treatment for removing organic pollutants, especially dyes.
  • Different type of AOPs such as photocatalysis, photolysis, UV/H2O2, photolysis, UV/O3, photo-Fenton, electrochemical oxidation, ozonation and sonolysis etc., can be effectively used for the treatment of dye containing wastewater.
  • This paper thoroughly investigated the current utilization of photolysis and photocatalytic treatment processes, which are among effective AOPs, for the degradation dye containing
  • The study confirmed that the photocatalysis could be used for the complete mineralization of various dyes present in water using light and a photocatalyst by the simultaneous occurrence of oxidation and reduction reactions.
  • pH, initial concentration of the dye, catalyst loading etc., were identified to have influence on the photocatalytic degradation of the dye.
  • A recent development in the photocatalytic process using TiO2 is the photocatalyst modification by metal and non-metal doping, which results in improved photocatalytic activity in the presence of visible radiation.
  • By utilizing a cost effective and sustainable energy source for the photocatalytic degradation of pollutants will be every effective.
  • Sunlight as an energy for the photocatalytic degradation of various dyes and other organic pollutants will be more efficient in terms of energy utilization.
  • The combination of various AOPs such as photocatalysis combined with Sonolysis, Ozonation, Electrolysis, Fenton etc., will be major aspects for the complete mineralization of various organic pollutants. It will be effective, as there is a synergy effect, by combining one or more AOPs and it will eliminate the drawbacks of individual processes.
  • Scaling up of an energy efficient, cost effective and sustainable technique for the complete mineralization of various type of organic pollutants from water by using the reusable energy source, sunlight and by individual AOPs or the hybrid AOPs is the challenging future aspect.

Reviewer 2 Report

The authors attempt to summarise recents investigations on the use of photolysis and photocatalysis treatment processes for the remediation of dye wastewater.

1- There exist many papers in this field. Moreover, similar review papers have already been reported by the other researchers. Please highlight more the novelty of this review paper.

2-Please add more paper on photocatalysis treatment. I suggest adding these references:   Journal of Photochemistry and Photobiology A: Chemistry 358, 111-120 (2018) ; Environmental Science and Pollution Research 26 (19), 19035-19046 (2019)

 3°) It would also better to add a part on contribution of reactives species (RSO) in oxidation process in order to enhance the quality of paper

4°) the part “Advanced oxidation processes (AOPs)” is very technical. It would be helpful to, when presenting the field of application (flow rate and concentration) of each process.

5°) Conclusions: what are the main take home messages (sustainable techniques, futures trends ??))

6°) Please add your viewpoint? Journal of ChemEngineering _MDPI readers expect from this paper a clear vision of the direction / treatment/ of these problems (dye water) in the future.

Author Response

Reviewer # 2 _ comments and our responses

Manuscript ID: ChemEngineering-1785338

Title: Current trends on the utilization of photolysis and photocatalysis treatment processes for the remediation of dye wastewater: A short review

  • There exist many papers in this field. Moreover, similar reviewpapers have already been reported by the other researchers.Please highlight more the novelty of this review paper.

Our responses:

The novelty of this review paper highlighted and incorporated into the text as “ Ensuring the availability of clean water is essential for humans, terrestrial and aquatic animals and plants. In compliance to the sustainable development goals (SDG) to ensure this availability in the pursuit of global economic growth and industrial developments, generated wastewater has to be treated and remediated for reuse. Textile industries commence one of the most labour intensive industries providing employment to various downstream and upstream sectors. At the same time, this industry produces a large amount of dye contaminated wastewater that is release into rivers and streams. The modern dyes are synthetically designed to withstand weathering processes or biodegradation. Over the last few decades, researchers and scientists have investigated new techniques and methods to treat and remediate dye contaminated wastewater. This review paper explores the current and updated trends in this field with the focus on photolysis and photocatalysis treatment process.  

  • Please add more paper on photocatalysis treatment. I suggest adding these references: Journal of Photochemistry and Photobiology A: Chemistry 358, 111-120 (2018) ; Environmental Science and Pollution Research 26 (19), 19035-19046 (2019)

Our responses:

Added as suggested:

MB    

Immobilised TiO2

[MB]0=75 mg/L,

[Zinc]= 60 mg/L, [NaCl]= 0.250 M, flowrate of 0.7 L/min.

After 180 min of UV-radiation 79.27% reduction of initial dye concentration was observed.

[104]

Add information’s related “Reaction kinetics model’ (please refer section 10) and new references (as suggested) also has been added. 

A mathematical model was studied under the heterogeneous photocatalytic degradation of wastewater containing petroleum [115]. The various parameters influencing the mechanism of degradation, such as mass transfer step, kinetics of the mineralization etc., were considered as the model parameters. Mass balances were chosen as bulk region and the catalyst phase was considered as the solid phase in order to develop the model. The degradation mechanism of solid phase was considered in two stages such as an equivalent intermediate (EI), in which the toluene is transformed to EI and in second stage, oxidation of EI gives carbon dioxide (CO2). The results found a good correlation between modeling and empirical data in terms of degradation and mineralization. The simulation of degradation kinetics of the target pollutant was obtained in the absence of reaction pathway. A good similarity with the experimental was obtained for the target pollutant from the model.

  • It would also better to add a part on contribution of reactives species (RSO) in oxidation process in order to enhance the quality of paper

Our responses:

Added a part on contribution of reactives species (RSO) in oxidation process as suggested and incorporated into the text as below. Kindly refer to the section 3. Advanced oxidation processes (AOPs)

“To identify the formation as well as reaction mechanism of reactive oxygen species (ROS) in each of the AOPs are found to be critical in studying the principles and the degradation mechanism of organic pollutants under various AOPs. The presence of ROS is termed as responsible for the effective mineralization of various organic pollutants under different AOPs. ROS can be produced by external energy sources such as photo, sono, electro etc., in the presence and absence of catalysts as well as secondary oxidants.”

  • the part “Advanced oxidation processes (AOPs)” is very technical. It would be helpful to, when presenting the field of application (flow rate and concentration) of each process.

Our responses:

Table 3 and table 4 presents the results of the degradation schemes presented and is relevant to the manuscript. It is our opinion that flow rate and concentration would be more relevant for a manuscript discussing about photoreactor modelling.

  • Conclusions: what are the main take home messages

Our responses:

The main idea is to promote photolysis and photocatalysis as a good option for dye wastewater remediation in practical application. The conclusion has been edited accordingly.

“One of the reasons of water pollution is the discharge of dye effluent into water bodies. Dye effluents in wastewater must be treated using efficient dye removal processes before being released into the environment as it is one of the causes of water pollution. It was confirmed that AOPs which facilitate the combinations of ultraviolet irradiation, catalysts and oxidants to produce hydroxyl radicals (OH) in solutions can be used as an effective technology for the degradation of dye contained wastewater.

  • AOPs are confirmed as a highly competitive technology in water treatment for removing organic pollutants, especially dyes.
  • Different type of AOPs such as photocatalysis, photolysis, UV/H2O2, photolysis, UV/O3, photo-Fenton, electrochemical oxidation, ozonation and sonolysis etc., can be effectively used for the treatment of dye containing wastewater.
  • This paper thoroughly investigated the current utilization of photolysis and photocatalytic treatment processes, which are among effective AOPs, for the degradation dye containing wastewater.
  • The study confirmed that the photocatalysis could be used for the complete mineralization of various dyes present in water using light and a photocatalyst by the simultaneous occurrence of oxidation and reduction reactions.
  • pH, initial concentration of the dye, catalyst loading etc., were identified to have influence on the photocatalytic degradation of the dye.
  • A recent development in the photocatalytic process using TiO2 is the photocatalyst modification by metal and non-metal doping, which results in improved photocatalytic activity in the presence of visible radiation.
  • By utilizing a cost effective and sustainable energy source for the photocatalytic degradation of pollutants will be every effective.
  • Sunlight as an energy for the photocatalytic degradation of various dyes and other organic pollutants will be more efficient in terms of energy utilization.
  • The combination of various AOPs such as photocatalysis combined with Sonolysis, Ozonation, Electrolysis, Fenton etc., will be major aspects for the complete mineralization of various organic pollutants. It will be effective, as there is a synergy effect, by combining one or more AOPs and it will eliminate the drawbacks of individual processes.
  • Scaling up of an energy efficient, cost effective and sustainable technique for the complete mineralization of various type of organic pollutants from water by using the reusable energy source, sunlight and by individual AOPs or the hybrid AOPs is the challenging future aspect.”

  • Please add your viewpoint? Journal of ChemEngineering_MDPI readers expect from this paper a clear vision of thedirection / treatment/ of these problems (dye water) in the future

Our responses:

Viewpoint  has been added, kindly refer to the conclusions sections

  • By utilizing a cost effective and sustainable energy source for the photocatalytic degradation of pollutants will be every effective.
  • Sunlight as an energy for the photocatalytic degradation of various dyes and other organic pollutants will be more efficient in terms of energy utilization.
  • The combination of various AOPs such as photocatalysis combined with Sonolysis, Ozonation, Electrolysis, Fenton etc., will be major aspects for the complete mineralization of various organic pollutants. It will be effective, as there is a synergy effect, by combining one or more AOPs and it will eliminate the drawbacks of individual processes.
  • Scaling up of an energy efficient, cost effective and sustainable technique for the complete mineralization of various type of organic pollutants from water by using the reusable energy source, sunlight and by individual AOPs or the hybrid AOPs is the challenging future aspect.

Round 2

Reviewer 2 Report

 I can recommend the Ms for publication now.